# LOWERING DATA DIVERSITY CAN ACCELERATE TRAINING: CASE STUDIES IN SYNTHETIC TASKS

## ABSTRACT

We identify a loss plateau at the start of training in the three synthetic settings of in-context linear regression, sparse parity, and fact memorization. While careful tweaks to the optimization algorithm can mitigate these plateaus, we find that a simpler orthogonal approach of *lowering the data diversity*, and in doing so, biasing the training distribution *away* from the test distribution, counter-intuitively also speeds up training. This connection between data diversity and training speed holds for three different diversity-*reducing* interventions across our varied synthetic settings. Our findings offer a new perspective on data filtering and curriculum learning for training machine learning models.

## 1  INTRODUCTION

How do we train neural networks faster, or avoid periods of slow progress during learning? The canonical answer points to extensive work on optimization, aiming to accelerate training via efficient optimizers for deep neural networks and theory for both convex and non-convex settings. In this work, we demonstrate that in a variety of settings, learning plateaus can be traced to issues with optimization. Our key finding is that this same effect of mitigating loss plateaus by better optimization can be achieved via a family of interventions on the *data distribution* that are *simple* and even *naive*, yet powerful.

We consider three synthetic tasks – in-context linear regression, sparse parity, and fact memorization – each marked by an initial phase of slow or plateaued learning, followed by abrupt learning to low loss. We begin by demonstrating that these plateaus can be avoided with suitable interventions on optimization: by biasing or removing interference between batch gradients we can mitigate the initial phase of slow learning.

The fact that tweaking the optimization algorithm suffices to mitigate these plateaus suggests it may be possible to develop more sophisticated optimization schemes that seamlessly handle such plateaus out-of-the-box. We take an orthogonal approach: we focus on the *data* being optimized over. We demonstrate that extremely simple yet astonishingly counter-intuitive *data* interventions can have a similar effect to complicated and subtle interventions on the optimization process.

While much work on the related topic of data filtering aims to find heuristics for high-quality data points that represent particularly salient or meaningful parts of the test distribution, our work is different in that we draw data *randomly* in our interventions, from distributions that are often *further* from the test distribution.

This is best illustrated with an example from our first synthetic setting: in-context learning of linear regression, where a Transformer is trained to take in sequences $[x_1, wx_1, \cdots, x_k, wx_k, x_{\text{query}}]$ for random $x_i$, infer the latent task vector $w$, and output a predicted regression estimate $\hat{y}_{\text{query}} = wx_{\text{query}}$. The task vectors $w$ are drawn from a specific distribution, and our goal is to learn the in-context ridge solution, which is the correct solution for Gaussian $w$. When training on this test distribution for $w$, the model eventually learns the optimal solution after an initial learning plateau. Figure 1 illustrates how *biasing the training distribution* by *reducing data diversity* (the number of task vectors we train with) can mitigate the learning plateau altogether. There is no free lunch, of course, as learning the optimal solution to the biased training distribution eventually does poorly on the test distribution of interest.

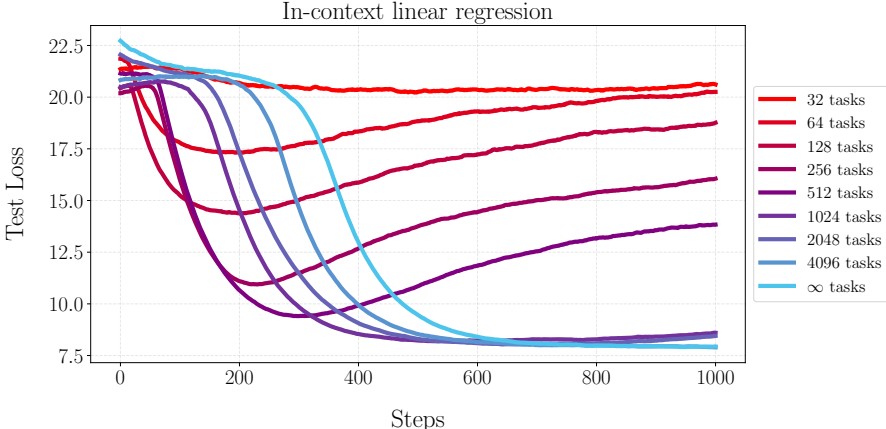

Figure 1: **Lowering task diversity can accelerate improvement on test loss.** In our in-context linear regression setup, the test distribution has an infinite set of tasks. We construct less diverse training distributions where we randomly subsample a fixed set of tasks. Surprisingly, training on these less diverse data distributions improve the test loss quicker at the start of training compared to training on the test distribution.

In this work, we present an associated family of such data interventions that have qualitatively similar effects. We show that *simple and counterintuitive data interventions* can match or outperform involved optimization interventions in accelerating training on a variety of synthetic tasks. This poses a tantalizing set of questions for future work on curricula, data filtering, optimization, and more, which we discuss in detail.

## 2 SETUP

In this section, we characterize the data distribution and training for the three synthetic settings we consider: in-context linear regression, sparse parity, and fact memorization. Given the range of our settings, we introduce each one with a short description and then a lengthier explanation of the setup. We provide a more detailed description below for those who are interested, though readers may feel comfortable skipping directly to Section 3 after reading the intuitive descriptions.

### 2.1 IN-CONTEXT LINEAR REGRESSION

In in-context linear regression, we train a transformer where each training *sample* is a sequence of $(x, y)$ points in $\mathbb{R}^d$, where $y = w^\top x + \epsilon$ for some noise $\epsilon$, and we train it to generalize to a new input at the end of the sequence and predict $y_{\text{query}}$ from $x_{\text{query}}$ by inferring $w$ in context.

**Data setup.** We are interested in learning functions $f \in \mathcal{F}$ that map inputs $x \in \mathbb{R}^d$ to outputs $y \in \mathbb{R}$. Our setup closely follows Kotha et al. (2024), focusing on linear regression for noisy data where every function is given by $f_w \colon x \mapsto \langle w, x \rangle$ for a fixed $w \in \mathbb{R}^d$. We are given a set of samples $S$ of variable length $k$ from 0 to maximum length $N$ such that

$$S = \{(x_1, y_1), \ldots, (x_k, y_k)\}, \tag{1}$$

with $y_i = f_w(x_i) + \epsilon_i$ and $\epsilon_i \sim \mathcal{N}(0, \sigma^2)$. From this, a model estimates the output $y_{\text{query}}$ for a given input $x_{\text{query}}$. We will refer to an instance from our function class $f_w$ as a *task*, and when it is clear from context, we will refer to tasks by the associated weight vector $w$. All inputs will be sampled from the normal distribution via $x_i \sim \mathcal{N}(0, I_d)$.

**Training.** We consider auto-regressive models $T_\theta$ that take in a sequence of tokens, each in $\mathbb{R}^d$, to produce a real-valued output. For samples $S$ generated under $w$ as in Equation 1, we feed $T_\theta$ the

*prompt* $[x_1, y_1, \ldots, x_k, y_k, x_{\text{query}}]^1$ and take its output as a prediction of $y_{\text{query}}$. When appropriate, we will refer to the $x_i$'s in the prompt as $X \in \mathbb{R}^{k \times d}$ and the $y_i$'s as $y \in \mathbb{R}^k$. We train and evaluate $T_\theta$ with respect to a weight distribution $\mathcal{D}$ via the quadratic loss. Further details in Appendix B.1.

$$\mathcal{L}(\theta, \mathcal{D}) = \sum_{k=0}^{N} \mathbb{E}_{\substack{w \sim \mathcal{D} \\ x_i \sim \mathcal{N}(0, I_d) \\ \epsilon_i \sim \mathcal{N}(0, \sigma^2)}} \left[ \left( T_\theta \left( [x_1, y_1, \ldots, x_k, y_k, x_{\text{query}}] \right) - y_{\text{query}} \right)^2 \right]. \tag{2}$$

by sampling a fresh batch of $x, w, \epsilon$ in each step. Under the quadratic loss, the optimal output is $\mathbb{E}\left[ f_w(x_{\text{query}}) + \epsilon \mid X, y \right] = \langle \mathbb{E}\left[ w \mid X, y \right], x_{\text{query}} \rangle$.

**Test distribution.** For this synthetic task, we will fix the test distribution to have each weight vector drawn from a Gaussian distribution, with weight $w \sim \mathcal{N}(0, I_d)$. In this case, the optimal solution is to perform *ridge regression*, or

$$w^*(X, y) = \mathbb{E}\left[ w \mid X, y \right] = \left( X^\top X + \frac{\sigma^2}{\tau^2} I_d \right)^{-1} X^\top y. \tag{3}$$

## 2.2 SPARSE PARITY

**Data Setup** Following prior work (Kearns, 1998; Barak et al., 2022), we consider inputs $x$ that are points on the $d$-dimensional hypercube $\{0, 1\}^d$ where the output is given by the product of $k$ bits $\chi_k(x) = \prod_{i=1}^{k} x_i$.

**Training** We train a 3-layer MLP to minimize the prediction loss under mean squared error when samples are drawn from $\mathcal{D}$.

$$\mathcal{L}(\theta, \mathcal{D}) = \mathbb{E}_{x \sim \mathcal{D}} \left[ (f_\theta(x) - \prod_{i=1}^{k} x_i)^2 \right]$$

**Test Distribution** For the test distribution, we will sample inputs uniformly from the $d$-dimensional hypercube. This constitutes a maximal diversity input distribution.

## 2.3 FACT MEMORIZATION

We train a transformer autoregressively to output the right *fact*, given an $(s, r)$ subject relation pair, with the added insertion of random noise tokens. To do the task properly it needs to memorize the facts and learn the structure of the objective, as well as learn to ignore the noise.

**Data Setup** We consider a setup designed to simulate learning facts from a text corpus (Ghosal et al., 2024). Following prior works on knowledge graphs (Petroni et al., 2019; Elsahar et al., 2018), we model facts as consisting of triplets of subject-entity, relation-type, and answer-answer entity. We construct a synthetic language with a set of subject tokens $\mathcal{S}$, relation tokens $\mathcal{R}$, answer tokens $\mathcal{A}$ and noise tokens $\mathcal{N}$ (i.e. the total token space $\mathcal{T} = \mathcal{S} \cup \mathcal{R} \cup \mathcal{A} \cup \mathcal{N}$). We assume that different relations induce a distinct set of plausible answers (i.e. $\mathcal{A} = \cup_{r \in \mathcal{R}} \mathcal{A}^r$) and that there is a predetermined ground-truth mapping $\phi((s, r)) \to a$ which determines the answer corresponding to every subject relation pair. We generate *documents*, $d$, by sampling a subject $s \sim \mathcal{S}$, a relation $r \sim \text{Unif}(\mathcal{R})$, and a set of $k_{\text{noise}}$ noise tokens sampled i.i.d. as $n_1, \ldots, n_{k_{\text{noise}}} \sim \text{Unif}(\mathcal{N})$. Then, the document is the token sequence

$$d = [s, r, n_1, \ldots, n_{k_{\text{noise}}}, \phi((s, r))] \tag{4}$$

Intuitively, the same underlying fact can be represented in multiple styles/formats in the pretraining corpus. We use the noise tokens to represent these axes of variation that are "irrelevant" to the knowledge graph structure. In this setup, we consider the learning of each subject $s$ as a separate task.

---

[1] Every 1-dimensional token is right-padded with $d - 1$ zeroes

**Training**    Similar to the ICL-LR setting, we study an autoregressive model $T_\theta$ mapping from sequences of tokens in the token-space $\mathcal{T}$ to produce a next-token prediction. We train with the standard auto-regressive language modelling objective on document $d = [s, r, n_1, ..., n_{k_{\text{noise}}}, \phi(s,r)]$, where $d_i$ denotes the $i$-th token of document $d$.

$$\mathcal{L}(\theta, \mathcal{D}) = \sum_{k=0}^{N} \mathop{\mathbb{E}}_{\substack{s \sim \mathcal{D} \\ r \sim \text{Unif}(\mathcal{R}) \\ n_i \sim \text{Unif}(\mathcal{N})}} [- \log T_\theta(d_{k+1} | d_0, ..., d_k)] \tag{5}$$

We detail model and hyperparameter implementation details in Appendix B.3.

**Test Distribution**    For the test distribution, in addition to sampling a relation and noise tokens, we uniformly sample a subject so that $s \sim \text{Unif}(\mathcal{S})$. We compute the factual memorization accuracy of our model by prompting with all but the last token in the document (i.e $[s, r, n_1, ..., n_{k_{\text{noise}}}]$) and checking for an exact match between the greedy-decoding and ground truth answer $\phi((s,r))$. In practice, we measure this accuracy by explicitly considering all subject relation pairs (with different noise tokens for each sample).

## 3    LOSS PLATEAUS AND HOW TO MITIGATE THEM VIA OPTIMIZATION

Many synthetic settings are affected by loss plateaus, where the model makes little to no progress on the loss and then has a phase of abrupt learning even when the train and test distribution are same. We find that in our settings, these plateaus are **not intrinsic**, and we can intervene on the gradients in the optimization procedure to bypass them and learn quickly. Our interventions are motivated by the intuition that interference is causing slow learning, where gradients from different data points and batches have low similarity and high variance, so the model makes overall slow progress. This hypothesis is difficult to check in all our settings due to the pitfalls of measuring variance when using adaptive optimizers, but we effectively use this intuition to speed up training.

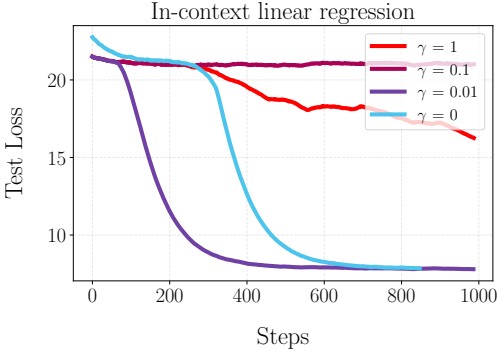
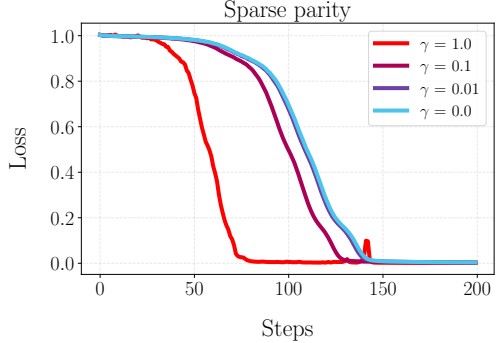

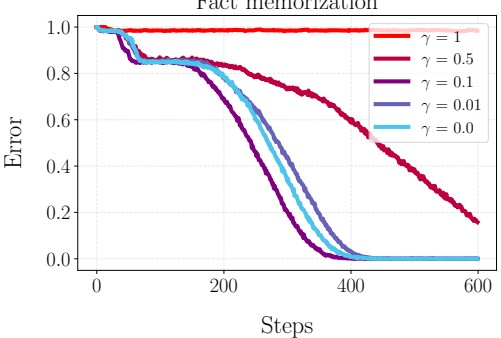

Figure 2: **Effect of biasing the gradients.** In this intervention, we induce a correlation between batch gradients by adding a small fraction $\gamma$ of a previous gradient to the current gradient. We find that this destabilizes training for high values of $\gamma$, while improving training for medium values of $\gamma$, before approaching standard training at $\gamma = 0$.

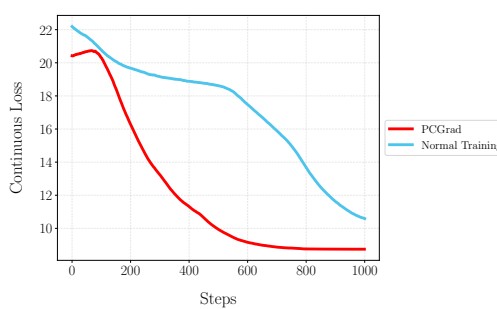

Figure 3: **Effect of PCGrad on training dynamics** We apply PCGrad, an algorithm designed to reduce gradient interference by projecting the gradients onto each other. We find that this results in faster improvement on a step-count basis, even though it is computationally slower.

**Biasing batch gradients.** We modify our training procedure to reduce gradient variance in the following way.

1. Throughout training, we maintain a gradient bias $g_{\text{bias}}$, which we store as the batch gradient at a given step, updated every $n$ steps (where $n = 100$ for in context linear regression and sparse parity, $n = 25$ for the facts setting).

2. For each step, we take the gradient $g_t$ at that step and instead take a gradient update with gradient $g_t + \gamma g'$, where $\gamma$ controls the extent to which we bias training.

In practice, this intervention helps speed up learning across our settings (Figure 2) for the right amount of biasing $\gamma$. Too high, and the model loses training signal, but too low and we recover normal training.

**PCGrad.** We also successfully apply PCGrad ((Yu et al., 2020)), an algorithm from multi task reinforcement learning that aims to make gradients of different samples orthogonal at each step of training.

1. For the loss on sample $i$ of each batch, compute its sample gradient $g_i$.

2. For each $g_i$, project it iteratively so it doesn't interfere with every other $g_j$, i.e. take a step with gradient $g_i^{PC} = g_i^{PC} - \frac{g_i^{PC} \cdot g_j}{||g_j||^2} g_j$

We apply this intervention to in-context linear regression. We find that this speeds up training in terms of the number of steps, despite the fact that we run it at a 4x lower batch size for memory and efficiency reasons (Figure 3). With less data and less steps, PCGrad induces a strong speedup relative to baseline. It is surprising that a technique designed for reinforcement learning works so well out of the box in this in context learning setting. This also validates our intuition that a phenomenon of interference is slowing down training, as PCGrad was built to solve for interference in RL. We discuss an additional intervention of training different layers with different learning rates in Figure 12.

## 4 LOWERING DIVERSITY CAN ACCELERATE TRAINING

In the previous section, we showed how the plateau can be removed by intervening on the optimization. We consider a **purely data-driven** approach to get the same effect, by lowering the *diversity* of the data. This may be reducing the interference described previously, enabling faster early-time progress, as this class of interventions also successfully removes the loss plateau:

- Decreasing the number of tasks or samples in the data distribution

- Sampling from the data with a randomly assigned power law instead of uniformly

- Increasing the correlation between tasks sampled (for in-context linear regression)

Our interventions have two surprising properties that defy traditional intuition about data selection.

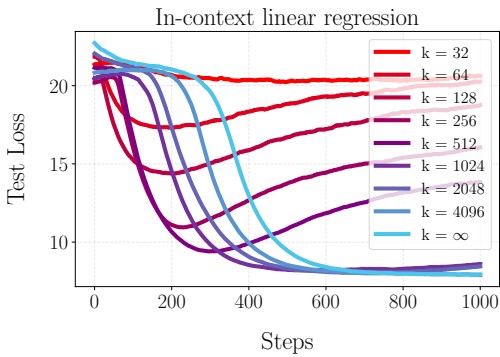
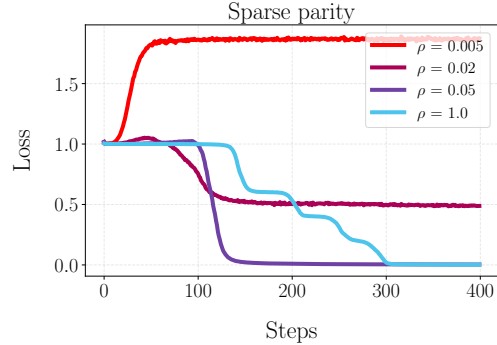

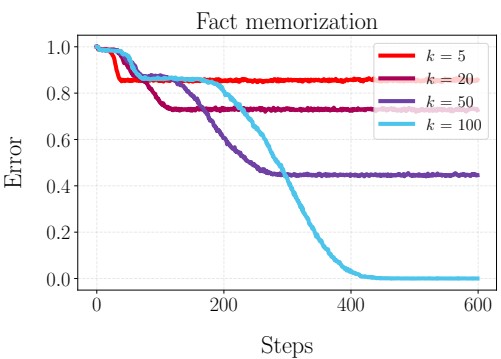

Figure 4: **Effect of varying the number of tasks.** For each setting, we randomly subsample the data. For in-context linear regression and fact memorization, we select $k$ tasks out of a total of $\infty$ and 100 tasks, respectively. For sparse parity, we select a random $\rho$ fraction of the data to be subsampled and run for $\frac{1}{\rho}$ epochs. We find that lowering the number of tasks increases early-time improvement at the cost of eventual memorization.

1. **Biasing away from the test distribution:** The training distributions we select are further from the original distribution used to measure test loss. The model is learning faster even when this gap is increased!

2. **Datapoint agnostic:** Our data selection strategies do not discriminate between datapoints and do not use any notion of quality or difficulty reminiscent of classical work on filtering and curriculum learning (refer to Section 5.1 for a detailed treatment).

In the following sections, we detail how our three interventions increase training speed across our synthetic settings. For all of the following training runs, we average over 5 seeds, except for linear regression experiments due to compute constraints.

## 4.1 LOWERING THE NUMBER OF TASKS OR SAMPLES

Our simplest intervention is randomly subsampling the data, either by subsampling tasks (when possible) or individual data points. We pick a subsample before training and then only take gradient steps on this data. Therefore, at the same number of steps, we will have seen less tasks, but trained on each one for more steps (Figure 4). We outline the exact subsampling mechanism for the three tasks below.

**Linear regression.** Instead of drawing every sample from the Gaussian $\mathcal{N}(0, I_d)$, we consider training over a "fixed" set of weights, sampling $w$ uniformly from $\mathbb{W}_k = \{w_1, \ldots, w_k\}$. Each $w_i$ is one sample from the true task distribution of $\mathcal{N}(0, I_d)$. We denote the discrete distribution we obtain $\mathcal{D}_k$. Though we recover the Gaussian distribution as $k \to \infty$, ridge regression is no longer optimal for finite $k$. The Bayes optimal estimator for $\mathcal{D}_k$ is:

$$w_k^*(X, y) = \frac{\sum_{w \in \mathbb{W}_k} w \varphi\left((y - Xw)/\sigma\right)}{\sum_{w \in \mathbb{W}_k} \varphi\left((y - Xw)/\sigma\right)} \qquad (6)$$

where $\varphi(\cdot)$ is the density of the standard multivariate normal distribution (derivation found in (Kotha et al., 2024)). This estimator effectively infers which of the $k$ tasks generated the data and then predicts for that one.

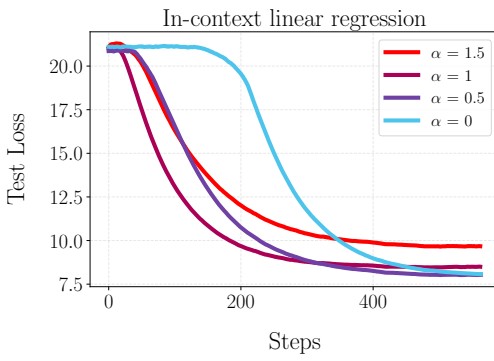
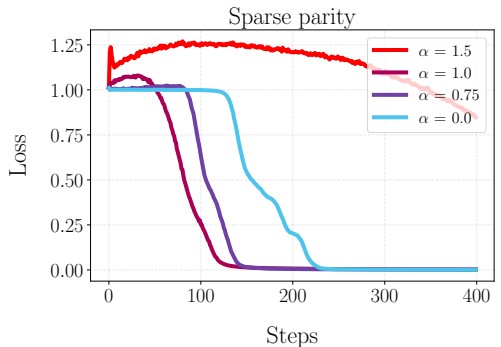

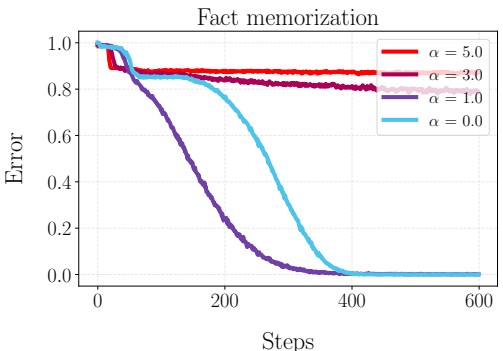

Figure 5: **Effect of varying the Zipf exponent.** For each distribution, we vary the rank frequency exponent, with $\alpha = 0$ being uniform and increasing $\alpha$ converging to a single task. We find that for a medium value of $\alpha$, there is a speedup over either extreme.

We observe (Figure 1) that for low task count, the model simply memorizes the finite set and never generalizes. For large enough task count the loss trajectory simply approaches that of training on infinite tasks. However, there is an intermediate regime where the model **initially makes progress to a generalizing solution** before eventually converging to a memorizing solution that does not generalize.

**Sparse parity.** We precompute a dataset $D$ of bit strings uniformly sampled from the hypercube, and train several MLPs on random subsets of varying size. The sizes are measured as fractions of the entire dataset, which in turn has its size matched to the number of steps times batch size. Each subset of fraction $\rho$ is traine on $1/\rho$ time so that the same number of points are seein in each case. Lower $\rho$ correspond to increasingly offline (many epoch) training, and we find that $\rho \ll 1$ results faster training, though as $\rho \to 0$ no learning takes place.

**Fact memorization.** In the fact-memorization setting, we test the impact of truncating the support of the subject distribution to $k$ entities. Concretely, during training we replace $\mathcal{D}$ with $\mathcal{D}_k = \{s_1, ..., s_k\}$, but leave the remainder of the data-generating process and the test distribution unmodified. In Figure 4, we observe that training on the truncated distribution mitigates the initial plateau in test-accuracy (measured on all subjects) that we observe when training on $\mathcal{D}$. However the models trained on the truncated distribution eventually stop improving in accuracy, simply because their training is omitting certain facts that must be memorized. Nevertheless, our findings replicate the intuition observed in the other settings: training on a less-diverse subset of the training distribution makes faster initial progress.

## 4.2 POWER LAW SAMPLING

We test other ways of modifying the sampling distribution and intervene on the data by sampling from the Zipf distribution rather than the uniform one. Given $k$ tasks/samples, we randomly sample each with probability proportional to $\frac{1}{i^\alpha}$ for frequency parameter $\alpha$ and randomly assigned index $i \in [k]$. Setting $\alpha = 0$ recovers the default uniform sampling distribution, while increasing $\alpha$ approaches

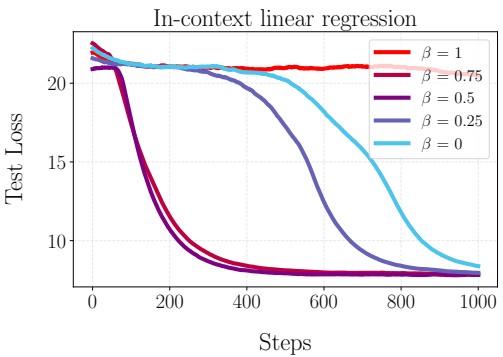

Figure 6: **Effect of biasing the covariance matrix.** In this intervention, we take the covariance matrix for in-context linear regression tasks and interpolate it with a rank 1 matrix. $\beta = 0$ corresponds to standard training, while $\beta = 1$ corresponds to minimally diverse data. Intermediate values of $\beta$ learn much faster than either extreme.

the minimally diverse distribution of a single task/sample. We observe that across settings there is an intermediate value of $\alpha$ where we recover this diversity speedup, shown in Figure 5.

**In-context linear regression.** For this setting, we first fix a large number of tasks from the Gaussian distribution so the dynamics would resemble the continuous case (Figure 1). Then, instead of uniformly sampling from this set of tasks to generate inputs (like Section 4.1), we sample according to the Zipf distribution. We get the fastest improvement in loss for $\alpha = 1$, with slower improvement when $\alpha$ is lower or higher.

**Sparse parity.** For a dataset of size $D$, instead of drawing batches uniformly at random, we draw them according to a Zipf distribution with parameter $\alpha$, and sweep $\alpha$, so that some batches are consistently drawn more often. It does not matter which batches are drawn repeatedly, just that some batches *are* drawn more repeatedly, to see faster training.

**Fact memorization.** We plot the fact memorization curves when subjects are sampled according to Zipf distribution. Concentrating on the early training setting, we observe that making the distribution more long tailed (i.e. increasing $\alpha$) speeds up the initial stage of learning. Later in training, values of $\alpha$ that are too high saturate or learn very slowly, similarly to when we truncate the subject distribution in Section 4.1. Interestingly, for $\alpha = 1$, we observe that facts are memorized significantly faster than the uniform case ($\alpha = 0$).

### 4.3 INCREASING TASK CORRELATION

In-context linear regression samples the tasks from a Gaussian with identity covariance, which means that each task dimension is independent. We can lower diversity in this setting by inducing arbitrary correlations between task dimensions. To do this, we sample a vector $v$ on the unit sphere and then interpolate between an identity covariance matrix and the rank 1 matrix $vv^\top$ such that

$$w \sim \mathcal{N}(0, (1 - \beta)I_d + \beta vv^T)$$

where $\beta$ controls the extent of the bias in the distribution. This approach is conceptually similar to our gradient biasing approach in Figure 2: instead of increasing the correlation between gradients, we increase the correlation between tasks, and this induces a similar effect (Figure 6). This particular task structure is not present in the other settings, so we only apply it to in-context linear regression.

### 4.4 SUMMARY OF DATA INTERVENTIONS

Across our data interventions, we observe that there is an "optimal" amount of biasing where initial learning is sped up relative to the baseline. Above this amount, the biasing will bring us too far from our original distribution such that the model simply fits the biased data instead of approaching the generalizing solution. Below this amount of biasing, we approach normal training without the intervention. This highlights a more general pattern across our interventions: lowering data diversity trades off correctly specifying the objective for faster early progress, inducing a speedup in an intermediate regime where the model can do both.

## 5 RELATED WORK

### 5.1 DATA SELECTION AND CURRICULUM LEARNING

A number of prior works study the benefits of curating or ordering the examples seen during training. Many prior works on data selection aim to identify a subset of high quality data by quantifying the contribution of individual data points, such as influence functions (Koh & Liang, 2020), datamodels (Ilyas et al., 2022; Engstrom et al., 2024), distance to margin (Sorscher et al., 2023), gradient metrics (Paul et al., 2023), data quality filters (Brown et al., 2020; Gadre et al., 2023; Li et al., 2024; Maini et al., 2024), and reweighting domains (Xie et al., 2023; Chen et al., 2023; Thrush et al., 2024). Another line of work instead intervenes on the *order* that data points are seen during training, typically in increasing order of difficulty. Hacohen & Weinshall (2019) obtains example difficulty scores from a pretrained model when training on image classification while Tay et al. (2019) uses example length as a proxy and applies curriculum learning in a reading comprehension task. (Wu et al., 2021) performs a meta-analysis finding that curricula are rarely helpful for training on real data. Pentina et al. (2014) examines curriculum learning in a multi-task setting by identifying shared information between tasks. Similarly, Zhang et al. (2020) uses a value function to identify tasks at the frontier of the policy's ability. In contrast to these settings, our data interventions improve training *without* discriminating between datapoints whatsoever.

### 5.2 SYNTHETIC SETTINGS AND ABRUPT LEARNING

Though we are the first to formalize this phenomenon, a number of prior synthetic settings show hints of lower diversity benefiting training. For example, related effects have been observed in meta-learning MNIST with a biased distribution (Kirsch et al. (2022), Figure 8), in-context learning digit recognition with a Zipf distribution (Chan et al. (2022), Figure 6c), in-context linear regression with subsampled tasks (Kotha et al., 2024; Raventós et al., 2023), and grokking modular arithmetic (He et al. (2024), Figure 3b). Reddy (2023) make a similar finding on the emergence of induction heads. Although these works also find that long-tailed data distributions induce in-context learning, they do not consider the impact of data on training speed, as we do. Moreover, we study and connect the impact of the training distribution on settings qualitatively different from in-context learning.

### 5.3 OPTIMIZATION INTERVENTIONS

Several works have focused on accelerating optimization for training. McCandlish et al. (2018) studies the impact of batch size, identifying that the signal to noise-ratio of gradients controls the appropriate choice of batch size. Jastrzebski et al. (2020); Faghri et al. (2020) examine the role of learning rate, finding that smaller learning rates can contribute to poor conditioning and higher levels of gradient noise. Singh et al. (2023); Raventós et al. (2023) examine the role of $\ell_2$ regularization specifically for in-context learning problems and find that the presence of regularization can induce in-context learning capabilities, even in the presence of lower task diversity or larger model capacity. Another related line of work has studied optimization challenges in multi-task learning. Schaul et al. (2019) finds that coupling between a policy and the data-generation process in reinforcement learning can lead to plateaus. Yu et al. (2020) observes that interference between gradients in multi-task learning can inhibit training and proposes a optimization intervention to correct this. Fu et al. (2024) aims to break loss plateaus by intervening on how optimization affects parameters. Collectively, these works demonstrate the significant role optimization parameters play in enabling efficient training. In this work, however, we showcase a *simple* intervention on the data with similar effects to some of these interventions in the settings we study.

## 6 DISCUSSION

In this work, we present a counter-intuitive finding: lowering the diversity of the training data by *simple random subsampling* can accelerate training for a general class of settings. Learning in these settings can also be sped up with a single optimization intervention, targeted at reducing interference. These results position data diversity as a rich way of understanding and controlling the optimization landscape without extra information about the individual samples of the distribution or complicated optimization interventions.

Our initial work here suggests there may be interesting connections between the data distribution and the optimization landscape of the model, due to the fact that intervening on them allows us to get the same speedup. Moreover, we observe that decreasing the task count in the in-context linear regression setting decreases gradient interference (Figure 8), which we measure by looking at the similarity of the model gradient on random, distinct batches. This supports the idea that these two levers are acting on some shared mechanism of interference. Exploring this connection in future work may enrich our understanding of training dynamics, abrupt learning, and the finegrained role of data distributional properties in training.

- **Diversity-reducing interventions work for real data:** If reducing diversity can improve training speed for real data, this would have exciting implications for many of the training runs we do today. For example, curriculum learning (Bengio et al., 2009) has faced little success for standard supervised learning (Wu et al., 2021). However, standard curricula typically use a heuristic notion of difficulty. The perspective of dataset diversity introduced in this work can inspire new, curriculum-like interventions going beyond these heuristics.

- **Diversity-reducing interventions do not work for real data:** It is a real possibility that the phenomena observed in this work does not transfer to real-world settings. This motivates a follow-up question: What property of real-world data makes it different from our synthetic tasks? It is possible that the correlations in real task distributions have a different structure, or that they already exhibit a power-law data distribution? For example, it is widely hypothesized that the frequency with which facts are observed during language model pretraining follows a power law distribution (Kandpal et al., 2023; Mallen et al., 2023). Similarly, prior work on in-context learning has hypothesized the role of the power-law distribution in its formation (Chan et al., 2022). Understanding the properties that diversity controls is an important direction for future work.

We hope that our observations can inspire future work in fundamentally understanding our synthetic settings and investigating the potential (or lack thereof) for data-driven training speedups.

## 7 LIMITATIONS

Our experiments are purely observational and although we provide some initial speculation and results, we do not claim to have strong understanding of the underlying phenomena resulting in these speedups. For example, we do not provide theoretical results concerning the nature of the speedup, but hope that this inspires interesting future work studying this phenomenon. Furthermore, we limit our analysis to three synthetic setups and it may not generalize to other machine learning problems. This limits the applicability of these findings, though there is a strong possibility these insights can inspire data-driven interventions in domains outside of these settings. Furthermore, though our experiments are consistent across settings, the conducted experiments can be broader across hyperparameters of the training algorithm, problem setting, and intervention. We hope that these initial results are taken as inspiration for understanding machine learning algorithms instead of ready-to-apply methods for today.

## 8 REPRODUCIBILITY STATEMENT

We will open-source the code necessary to recreate these experiments to encourage future research on understanding the fundamental principle behind the speedup.

## 9 ETHICS STATEMENT

In this work, we work on generically improving the speed of training machine learning models. Our result currently only applied to synthetic settings with little practical impact, though the impact would be similar to any generic optimization routine in terms of training speed. Lowering the diversity of the data might potentially lower the coverage of under-represented groups in real data, which merits future research.

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

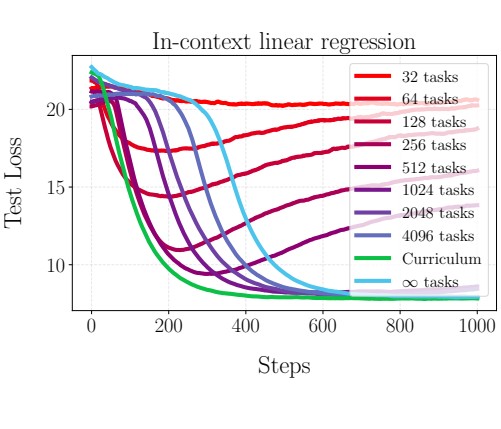

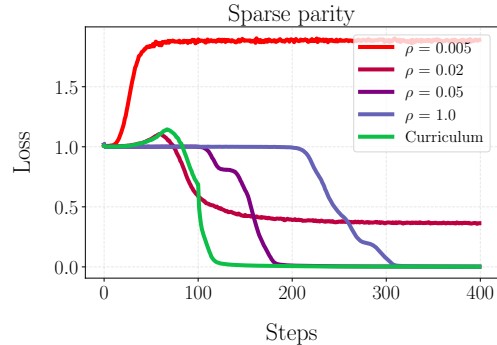

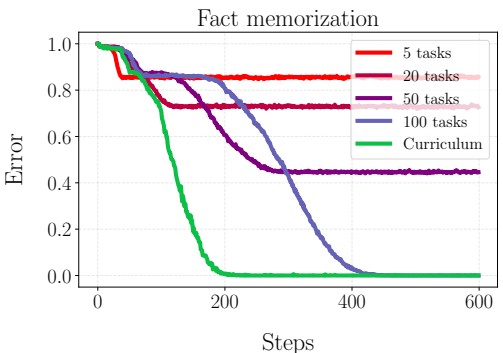

Figure 7: **Curriculum learning over number of tasks/samples.** For each setting, we vary the proportion of data subsampled over the course of training, increasing the diversity over the course of training. We find that using such a curriculum can achieve the best of both worlds, with faster test loss reduction early in training with convergence to optimal test loss at the end of training.

## A    CURRICULUM LEARNING EXPERIMENTS

For each setting, we introduce a curriculum over the number of subsampled tasks/samples, where we progressively increase the diversity of the distribution over the course of training. We implement this in the following way across our different settings

**In-context linear regression.**    We start with 32 tasks for 30 steps and double the number of tasks every 20 steps. When we double the number of tasks from $k$ to $2k$, we retain the original $k$ tasks and add $k$ new ones.

**Sparse parity.**    We start with 2% data diversity, or $\rho = 0.02$, until 100 steps, at which point we switch to all the data or $= 1.0$.

**Factuality.**    We start with $k = 20$ tasks until 100 steps, at which point we switch to the full set of $k = 100$ tasks for the remainder of training.

**Results.**    We find that this intervention is competitive or better than any other subsampling strategy at every step count, indicating that regardless of the compute budget, it is best to use this curriculum. This indicates we can achieve the best of both worlds–fast test loss improvement at the start of training, and a better generalizing solution by the end of training.

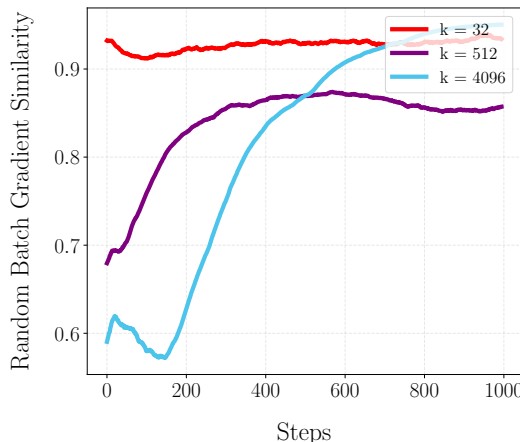

Figure 8: **Cosine similarity of model gradients on random batches, averaged over seeds**. Note that at a given run this measure is very noisy across runs and can quickly vary from $-0.5$ to $.5$, suggesting the instability in this early training.

## B    EXPERIMENTAL SETUP

### B.1    IN CONTEXT LINEAR REGRESSION DETAILS

We train 22.4M parameter GPT2 family models with an embedding dimension of 256, 12 layers and 8 attention heads. We use the Adam optimizer with $\beta_1 = 0.9, \beta_2 = 0.999$, a learning rate of $1e - 4$, and a batch size of 128.

To generate our linear regression data we sample 41 20-dimensional points per sequence, and use a noise std $\sigma^2 = 1$.

### B.2    SPARSE PARITY DETAILS

We take 50000 steps of training on sparse parity inputs of dimension 20, where the product of 6 bits defines the final answer. We train a three layer MLP with hidden layers of dimension 40, mapping $\mathbb{R}^{20} \rightarrow \mathbb{R}^1$. We train with SGD using learning rate 0.1 and batch size 20.

### B.3    FACTUALITY SETTING DETAILS

We train a small transformer with 6 layers and 6 heads and embedding dimension 192. We train with a learning rate of 5e-5 and a batch size of 64. We use the Adam optimizer with $\beta_1 = 0.9, \beta_2 = 0.95$.

## C    MISC

### C.1    TRACKING COSINE SIMILARITY OF GRADIENTS FOR DIFFERENT TASK COUNTS

To check whether gradient interference is related to lowering diversity, we track the cosine similarity of gradients over the course of training with different task counts. In line with expectations, we see that for higher task count, the cosine similarity is much lower, while for lower task counts, all the gradients are pointing in similar directions. This shows observational evidence that reducing the tasks may be improving optimization by reducing gradient interference.

### C.2    SCALING THE BATCH SIZE WITH THE NUMBER OF TASKS PRESERVES THE SPEEDUP

If the phenomenon at play within the setting of in-context linear regression is one of interference across tasks, it makes sense to check whether the effect persists if you scale the batch size propor-

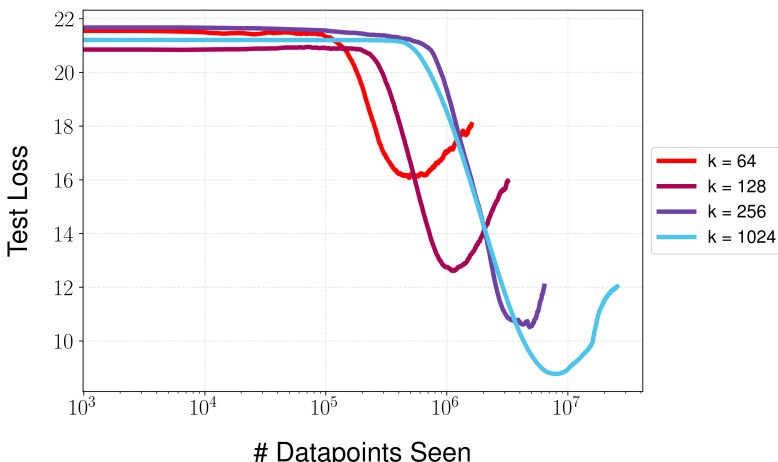

Figure 9: **Learning curves varying k in in context linear regression, where we scale batch size with k.** We check if our speedup persists if we scale the batch size $B$ with k, keeping $\frac{k}{B}$ constant. We report the loss in terms of the number of datapoints seen for a fair comparison, and see that lower k indeed still does better.

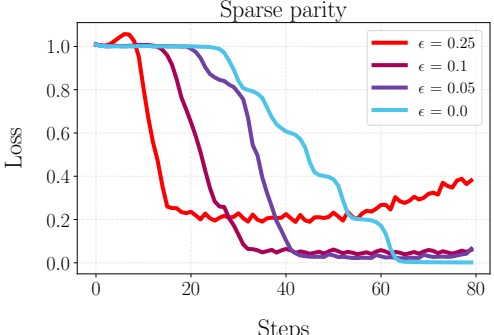

Figure 10: **Staircase intervention for sparse parity.** Training on approximate staircase functions $\chi_k + \epsilon \sum_{j<k} \chi_j$ instead of the test function $\chi_k$ accelerates learning of the test function $\chi_k$.

tionally to the number of tasks. This gives us the opportunity to check for a given batch size whether the interference is just related to having more tasks to sample from within a batch.

We observe in figure 9 that scaling the batch size proportionally with the number of tasks we subsample does not reverse this low diversity speedup, highlighting that there is a richer form of interference at play.

### C.3 STAIRCASE INTERVENTION FOR SPARSE PARITY

Taking inspiration from the work of (Abbe et al., 2021), we hypothesize that biasing the training distribution towards the staircase function $S_k = \sum_j \chi_j$ while downweighting the coefficients of all monomials $\chi_j$ by a factor $\epsilon$ allows us to "climb the staircase" to learn the monomial faster, while incurring only a small penalty for learning lower-order coefficients with power $\epsilon$. Formally, we train on $\chi_k + \sum_{j<k} \epsilon \chi_j$, and find there exists a range of $\epsilon > 0$ such that this reaches low loss faster than training on just $\chi_k$, with learning curves/speedups qualitatively resembling those of other interventions.

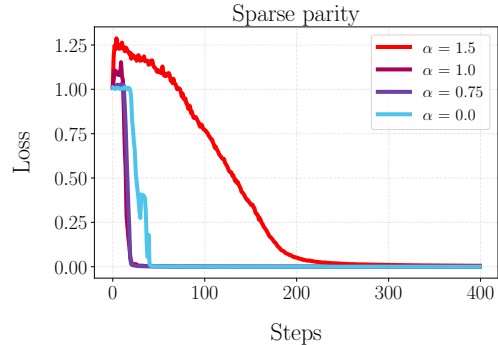

Figure 11: **Maximal learning rate training for the Zipf distribution for sparse parity.** We push the learning rate to the maximum value without instability and find that the speedup persists in this setting.

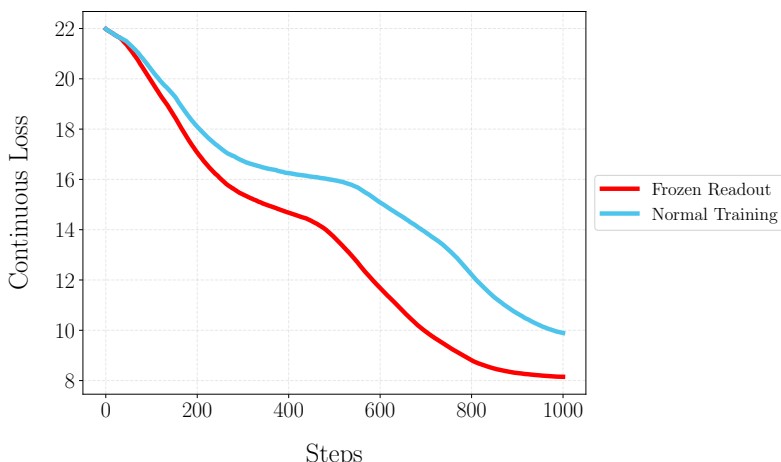

Figure 12: **Training dynamics when training with the readout frozen**

## C.4 MAXIMAL LEARNING RATE FOR SPARSE PARITY ZIPF

To check that our effects are not a product of badly tuned hyperparameters, we push the learning rate up to its maximal stable value of 0.1, and see that $\alpha = 0.75$ still learns faster, averaged over seeds, than $\alpha = 0$.

## C.5 FREEZING THE READOUT LAYER FOR IN-CONTEXT LINEAR REGRESSION

In the in-context learning setting, we noticed that when we decomposed the similarity of the random gradients (mentioned in Appendix B.3) in terms the model layers (simply separating out the normalized dot product terms in the sum), most of the noise and spiking concentrated in the readout layer that projects to the model output.

So we did an intervention where we train with this layer frozen, and do observe a moderate speedup, although less pronounced than in our other settings.

