# OpenReview forum: "Lowering Data Diversity can Accelerate Training: Case Studies in Synthetic Tasks"
_ICLR.cc/2025/Conference — Submitted to ICLR 2025_

### Official Review · Reviewer_7o5W · 2024-10-25

**Soundness:** 3
**Presentation:** 3
**Contribution:** 2
**Rating:** 5
**Confidence:** 4

**Summary:**

This paper examines methods to accelerate training by reducing data diversity, focusing on synthetic tasks like in-context linear regression, sparse parity, and fact memorization. Traditional approaches typically improve training by adjusting optimization algorithms to mitigate plateaus; however, this study found that simple interventions in data sampling, such as reducing task diversity or sampling with non-uniform distributions, can achieve similar benefits. These findings provide insights into data filtering and curriculum learning approaches, suggesting that less diverse but strategically chosen training data could enhance model efficiency.

**Strengths:**

The paper presents a clear experimental setup, demonstrating how intentional data biases can speed up learning in synthetic tasks.

The use of visuals makes its findings easier to understand.

This research is useful in situations where training efficiency is a priority.

**Weaknesses:**

The application scope is narrow to meet real-world needs, and it lacks an explanation for the observed phenomena.

**Questions:**

Your approach and optimization algorithms can achieve the same goal, so how would you convince others to use your method instead of an optimization algorithm?

Would combining your approach with other optimization algorithms lead to more significant performance improvements?

---

> ### Author Response · Authors · 2024-11-21
>
> Thank you for your helpful review! Hopefully, the following comments address your main concerns about our paper.
>
> > The application scope is narrow to meet real-world needs, and it lacks an explanation for the observed phenomena.
>
> We agree that a stronger understanding or real-world replication of such effects would make a much more compelling paper. However, as discussed in our general response, we believe that identifying interesting empirical phenomena in synthetic settings is really valuable for future research. For example, the original in-context linear regression paper (https://arxiv.org/abs/2208.01066) has no such theory or real-world experiments but has inspired dozens of impactful theorems and real-world studies; we find this to be a feature of several other impactful papers such as https://arxiv.org/abs/2306.00946, https://arxiv.org/abs/2306.15063, https://arxiv.org/abs/2405.15071, https://arxiv.org/abs/2301.05217. We hope that our empirical results can serve as similar inspiration for future work in fine-grained understanding or practical applications.
>
> > Your approach and optimization algorithms can achieve the same goal, so how would you convince others to use your method instead of an optimization algorithm?
>
> We would like to clarify that our goal is to not purpose this as a general speedup method thats ready for practice; instead, our consistent observation in these settings about the effect of data on optimization is surprising and will hopefully motivate more general interventions in the future. Additionally, many optimization algorithms are computationally inefficient or not suited for modern hardware. For example, using PCGrad is not scalable in general as it has O(batch size^2) complexity at each step.

---

### Official Review · Reviewer_yRQX · 2024-11-03

**Soundness:** 1
**Presentation:** 2
**Contribution:** 1
**Rating:** 3
**Confidence:** 4

**Summary:**

This paper argues that convergence using SGD can be more efficiently obtained, in some cases, by using a training data distribution that has less in common with the test distribution. This argument is of course counterintuitive, and it disagrees with a wide body of fundamental research in the field around how an ideal training dataset is one that has more in common with the general distribution, not less.

**Strengths:**

The authors take a stance that, to my knowledge, has not been studied before in machine learning. If this were a well understood, real phenomenon, I think it could be impactful.

**Weaknesses:**

Overall I found this paper’s argument unconvincing, as they experiment only on three synthetic settings and provide no arguments as to why this effect might occur. Of course, this effect cannot be shown forever, and eventually this skewing of the training distribution will have a negative effect on test performance. When and why does this happen, and what can be gleaned from this work about more conventional model training?

As the authors write themselves in the limitations section, they do not have an understanding of when or why this occurs, and do very limited evaluation and show preliminary results. I realize this section was written to get ahead of criticism they anticipate to this effect, but the work seems very incomplete with these questions left unanswered. The future work in the discussion section, for example, is very much in-scope, and should be part of the paper the authors are writing.

At this time, I feel that the presented investigation is not thorough enough to meet the bar for publication in ICLR.

**Questions:**

1. There is a wide body of work on generalization through a theoretical lens, typically taking advantage of both model complexity and train/test distribution similarity. How does this work fit into this literature?

2. Does this happen in more conventional language modeling (or non language modeling tasks)?

3. Why does this effect happen at all? I don't see any justification of this phenomenon, only a few demonstrations of it.

---

> ### Author Response · Authors · 2024-11-21
>
> Thank you for your helpful review! Hopefully, the following comments address your main concerns about our paper.
>
> > Of course, this effect cannot be shown forever, and eventually this skewing of the training distribution will have a negative effect on test performance.
>
> We introduce a curriculum learning method (detailed in Appendix A) where we slowly increase the diversity of the data over time. In all three of our settings, this curriculum uniformly dominates training on every other diversity level. Namely, for every step count, training via the curriculum achieves lower test loss compared to training on any fixed task/sample count, including the original test distribution. As such, our insight on lowering diversity can be used not only to speed up the start of training but can also be used to find the best generalizing test solution by the end of training.
>
> > Overall I found this paper’s argument unconvincing, as they experiment only on three synthetic settings and provide no arguments as to why this effect might occur. … When and why does this happen, and what can be gleaned from this work about more conventional model training?
>
> We agree that a stronger understanding or real-world replication of such effects would be highly interesting and impactful. However, as discussed in our general response, we believe that identifying interesting empirical phenomena in synthetic settings is a valuable step towards this, and the connections we suggest between optimization and data will inform this future work. Empirical results on synthetic tasks have previously played an important role in spurring research into other machine learning phenomena. For example, the original in-context linear regression paper (https://arxiv.org/abs/2208.01066) has no such theory or real-world experiments but has inspired dozens of impactful theorems and real-world studies; we find this to be a feature of several other impactful papers such as https://arxiv.org/abs/2306.00946, https://arxiv.org/abs/2306.15063, https://arxiv.org/abs/2405.15071, https://arxiv.org/abs/2301.05217. We hope that our empirical results can serve as similar inspirations for future work in fine-grained understanding or practical applications.
>
> > There is a wide body of work on generalization through a theoretical lens, typically taking advantage of both model complexity and train/test distribution similarity. How does this work fit into this literature?
>
> There are many interesting connections between our work and prior theoretical work on generalization. For example, our work connects to the ability to find generalizing solutions with underspecified data in the study of simplicity bias (https://arxiv.org/abs/1806.08734, https://arxiv.org/abs/1805.08522, https://arxiv.org/abs/2211.11567, https://arxiv.org/abs/2402.04362,). Though prior work studies how the model evolves from simpler solutions to more complex solutions (i.e. ridge regression to discrete regression), our work finds that the speed of training is different depending on the diversity of the data distribution. There is also a list of connections we discuss between our work and theoretical studies on optimization, curriculum, and synthetic settings in our related work.
>
> > Does this happen in more conventional language modeling (or non language modeling tasks)?
>
> We have not tested this ourselves. However, other papers test the setting of repeating the data points for conventional language models (https://arxiv.org/abs/2305.16264). In this setting, there does not appear to be a speedup in test loss convergence at the start of training. As mentioned in our discussion, we would be excited to either see the success of decreasing diversity or understanding why real data does not benefit from this.
>
> > Why does this effect happen at all? I don't see any justification of this phenomenon, only a few demonstrations of it.
>
> We were also quite surprised by this effect and want to share it with the broader research community to inspire explanations. We have some initial intuition that these landscapes serve as a challenge for basic optimization techniques, evidenced by the success of gradient biasing and PCGrad. We believe this may be connected to why reducing data diversity helps performance, and the lack of a complete justification contributes to the need to share our findings, rather than waiting for a full explanation.

---

> > ### Comment · Reviewer_yRQX · 2024-12-02
> > **response**
> >
> > It seems that the authors agree with my criticism. I'm electing to keep my score, as I believe these questions need to be answered in order for the paper to meet the standards of this venue.

---

### Official Review · Reviewer_A9cn · 2024-11-04

**Soundness:** 2
**Presentation:** 1
**Contribution:** 1
**Rating:** 1
**Confidence:** 4

**Summary:**

The paper demonstrates with experiments that lowering training data diversity accelerates training. In 3 experiment setups with synthetic data---in-context linear regression in transformers, learning sparse parity functions with MLP, and fact memorization in transformers---the authors argue that a variety of interventions can accelerate training (e.g. interventions on the gradient updates or on the training dataset).

**Strengths:**

The paper presents experiments on different setups, involving both transformers and MLPs that are used in practice. There has been recent interest in using transformers/MLPs to learn sparse parities [1, 2] and in-context linear regression [3, 4], which the paper studies. The general problem in the paper: of accelerating training is of interest to the ML community. The paper takes approaches this problem from the lens of data curation goes back to Curriculum Learning [5].

[1] Barak, B., Edelman, B., Goel, S., Kakade, S., Malach, E., & Zhang, C. (2022). Hidden progress in deep learning: Sgd learns parities near the computational limit. Advances in Neural Information Processing Systems, 35, 21750-21764.

[2] Edelman, B. L., Goel, S., Kakade, S., & Zhang, C. (2022, June). Inductive biases and variable creation in self-attention mechanisms. In International Conference on Machine Learning (pp. 5793-5831). PMLR.

[3] Garg, S., Tsipras, D., Liang, P. S., & Valiant, G. (2022). What can transformers learn in-context? a case study of simple function classes. Advances in Neural Information Processing Systems, 35, 30583-30598.

[4] Zhang, R., Frei, S., & Bartlett, P. L. (2023). Trained transformers learn linear models in-context. arXiv preprint arXiv:2306.09927.

[5] Bengio, Y., Louradour, J., Collobert, R., & Weston, J. (2009, June). Curriculum learning. In Proceedings of the 26th annual international conference on machine learning (pp. 41-48).

**Weaknesses:**

# Motivation
The paper aims to accelerate training for the sake of optimization only, but that is not the central goal in learning. In fact, research in ML optimization techniques serve to fulfill the **central goal of generalizing to the test distribution**. Moreover, the paper frequently finds in their experiments that faster training (using the methods they present) often leads to worse generalization, so I doubt the benefits of the proposed methods.

**In fact, the authors mention this exact severe limitation in line 52.** I'd find insights into generalization more helpful than just focusing on training optimization.

## On reducing data diversity to optimize faster
It is not shown that accelerating training by reducing data diversity leads to increased generalization to the test distribution. I think that would be a good argument for "reduce data diversity -> thus train faster -> thus generalize to test distribution better". As an example, consider the setting where we want to learn the target function $f(x) = x^2$ given a finite number of training samples $x_1, ..., x_n$. We can forget about the target (test distribution) and instead sample from the function $g(x) = 1$, _just for the sake of low data diversity_. Optimizing on samples to learn $g$ is faster, but this is far from the goal of learning the target $f$.

On another note, decreasing the number of training tasks/samples to reduce training data diversity should intuitively lead to overfitting. Figure 4 shows this exact phenomena, so I'm not convinced that accelerated training with the paper's data curation methods is actually helping in learning.

# Techniques
In Section 3, the paper argues that biasing batch gradients can accelerate training. Isn't this technique the same as SGD with momentum or an adaptive gradient method like AdaGrad or Adam? I'm not sure this is a novel contribution.

Finally, the paper dos not conduct data-diversity experiments on real-world problems such as image classification (even on well-studied datasets like CIFAR or MNIST) and language generation. I appreciate that the authors mention this limitation in Section 6 and 7, but I believe the current version of the manuscript is severely lacking with just synthetic data experiments.

# Writing
The paper needs numerous citations in the introduction to substantiate claims. Several typos and clarifications needed:
1. Citations for lines 41-44.
2. Line 96. It's verbose to call the task $f_w$ when we can simply write $w^T x + \epsilon$ as used in Line 93. Lines 96-105 can be written in 3-5 lines at max.
3. Line 297. The paper conflates "faster optimization" with "learning". The model trains faster, but does not generalize (or "learn") better or faster.

Moreover, the statement "low data diversity accelerates training" is repeated throughout the paper (paraphrased in many ways), making the paper more verbose than necessary. I understand that it is the paper's main argument, but it is overly used in my view.

**Questions:**

# High-level questions
1. Section 2.2. There is a notion of "task" in linear regression and fact memorization problems. What is a task in the sparse parity problem?
2. Section 2.2. Why study 3-layer MLP here but transformers for the other two problems?
3. Line 165. What is the $i^th$ token $d_i$? If these are the elements $s, r, n_1, ..., n_{k_noise}$, it doesn't make sense to have the transformer predict noise token $n_t$ given $s, r, n_1, ..., n_{t-1}$ since the noise tokens are drawn independently. If I'm reading this wrong, what is the tokenization procedure here?

# Low-level questions
1. Line 32: "followed by abrupt learning to low loss". What does this question mean? Are the authors pointing to the "grokking" line of work [1]?
2. Line 93. Typo: $x$ is in $\mathbb{R}^d$ but $(x, y)$ is not.
3. Equation 3. What is $\tau$?
4. Line 132. What is $\mathcal{D}$ here? Uniform distribution over the $d$-dimensional hypercube?
5. Line 162. What is the ICL-LR setting? It is likely the 1st problem, but this abbreviation is introduced here for the first time.

[1] Power, A., Burda, Y., Edwards, H., Babuschkin, I., & Misra, V. (2022). Grokking: Generalization beyond overfitting on small algorithmic datasets. arXiv preprint arXiv:2201.02177.

---

> ### Author Response · Authors · 2024-11-21
>
> Thank you for your helpful review! Hopefully, the following comments address your main concerns about our paper.
>
> We would like to start with an important clarification that should address a majority of the criticism. In all of our plots, we evaluate loss on the test distribution, not the train distribution. For example, even if we train on 256 tasks instead of infinite tasks, we evaluate the loss on the infinite task distribution. Therefore, we are achieving generalization/learning faster, and are in line with “reduce data diversity -> thus train faster -> thus generalize to test distribution better”. We believe this should ease a majority of the technical concerns about this paper, and we hope that you find the result as surprising/counter-intuitive as we did. Below, we address individual comments.
>
> > On another note, decreasing the number of training tasks/samples to reduce training data diversity should intuitively lead to overfitting. Figure 4 shows this exact phenomena, so I'm not convinced that accelerated training with the paper's data curation methods is actually helping in learning.
>
> We agree that in certain settings (i.e. factuality), training on less diverse data can result in a suboptimal solution for infinite training steps. In light of this, we introduce a curriculum learning method (detailed in Appendix A) where we slowly increase the diversity of the data over time. In all three of our settings, this curriculum uniformly dominates training on every other diversity level. Namely, for every step count, training via the curriculum achieves lower test loss compared to training on any fixed task/sample count, including the original test distribution. As such, our insight on lowering diversity can be used not only to speed up the start of training but can also be used to find the best generalizing test solution by the end of training.
>
> > In Section 3, the paper argues that biasing batch gradients can accelerate training. Isn't this technique the same as SGD with momentum or an adaptive gradient method like AdaGrad or Adam? I'm not sure this is a novel contribution.
>
> We agree, we do not claim that the gradient biasing and PCGrad experiments are novel. However it's important to note that our experiments are using Adam, an adaptive optimizer, so it seems that biasing in this way has a more pronounced effect. These gradient experiments are discussed to motivate our experiments on data and potential explanations for why changing the data distribution affects the speed of training.
>
> > Finally, the paper does not conduct data-diversity experiments on real-world problems such as image classification (even on well-studied datasets like CIFAR or MNIST) and language generation. I appreciate that the authors mention this limitation in Section 6 and 7, but I believe the current version of the manuscript is severely lacking with just synthetic data experiments.
>
> We agree that a stronger understanding or real-world replication of such effects would make a much more compelling paper. However, as discussed in our general response, we believe that identifying interesting empirical phenomena in synthetic settings is really valuable for future research. For example, the original in-context linear regression paper (https://arxiv.org/abs/2208.01066) has no such theory or real-world experiments but has inspired dozens of impactful theorems and real-world studies; we find this to be a feature of several other impactful papers such as https://arxiv.org/abs/2306.00946, https://arxiv.org/abs/2306.15063, https://arxiv.org/abs/2405.15071, https://arxiv.org/abs/2301.05217. We hope that our empirical results can serve as similar inspirations for future work in fine-grained understanding or practical applications.

---

> ### Author Response · Authors · 2024-11-21
>
> > There is a notion of "task" in linear regression and fact memorization problems. What is a task in the sparse parity problem?
>
> We note that sparse parity does not have explicit task structure. Our notion of data diversity can operate over task space (like linear regression and factuality) or sample space. When we apply the subsampling and power law interventions for sparse parity, we utilize diversity over samples instead of tasks.
>
> > Why study 3-layer MLP here but transformers for the other two problems?
>
> We wanted to cover a range of qualitatively different settings to test the robustness of our conclusions. We note that our results hold for the architectures corresponding to the original paper introducing the synthetic settings, which ends up being 3-layer MLPs in sparse parity and GPT-2 style transformers in factuality and linear regression. This reinforces the generality of our phenomenon: it holds across architectures and types of learning problems.
>
> > it doesn't make sense to have the transformer predict noise token
>
> The noise tokens are a synthetic structure we use to mimic how in real language, there are many tokens that are a number of tokens that are irrelevant to the task at hand.
>
> > "followed by abrupt learning to low loss
>
> We are referring to our results where we find that training on the original data has an initial loss  plateau followed by a quick decrease in loss–we will be more clear about this.
>
> Thank you for identifying our notation errors: $\tau$ is meant to be $1$ here in proportion to the variance of the weight prior, $\mathcal{D}$ is in fact the hypercube, and ICL-LR is meant to stand for in-context linear regression. We appreciate all the other writing advice, which we will definitely incorporate this in a camera-ready version of this work.

---

> > ### Comment · Reviewer_A9cn · 2024-11-22
> > **Reply to Rebuttal #2**
> >
> > Thank you for clarifying my questions about the notion of task in the parity problem, and studying 3-layer MLPs for them.
> >
> > > The noise tokens are a synthetic structure we use to mimic how in real language, there are many tokens that are a number of tokens that are irrelevant to the task at hand.
> >
> > I am not sure I understand.
> >
> > > We are referring to our results where we find that training on the original data has an initial loss plateau followed by a quick decrease in loss
> >
> > How are your results different from Grokking: https://arxiv.org/abs/2201.02177 ?
> >
> > In summary, **I am not sure this manuscript is ready for the conference at the moment, given limitations in motivations and experimental settings**. Thinking about data diversity is curious, however, and I'd be interested in seeing future work on how it affects generalization properties.

---

> ### Comment · Reviewer_A9cn · 2024-11-22
> **Reply to Rebuttal #1**
>
> I appreciate that the authors replied with clarifications. I'll respond to their comments below:
>
> > Therefore, we are achieving generalization/learning faster, and are in line with “reduce data diversity -> thus train faster -> thus generalize to test distribution better”.
>
> I think the authors miss a subtle point I make here. By "generalize to test distribution better" I do not mean "faster", but actually lower test loss after sufficiently many iterations of training. I did observe that the manuscript reports test loss on the y axis for all datasets, hence measuring generalization. However, in scenarios with high train-test distribution mismatch (small # tasks in Figure 1, small $k$ or $\rho$ in Figure 4), the test error plateaus at higher test loss values compared to scenarios with low mismatch (large # tasks1 in Figure 1, large $k$ or $\rho$ in Figure 4). These plots tell me that reducing train data diversity leads to worse generalization on test loss. Reviewer (2yyz) also mentions overfitting---I reiterate that the manuscript is essentially discovering overfitting in settings of low train data diversity. I am not sure that this is a counter-intuitive observation.
>
> > In light of this, we introduce a curriculum learning method (detailed in Appendix A) where we slowly increase the diversity of the data over time. In all three of our settings, this curriculum uniformly dominates training on every other diversity level.
>
> Thank you for running these experiments. These results are more in line with my expectations, since there is a long line of curriculum learning where training on increasingly complex tasks leads to both faster and better generalization. However, I do not think that these results are novel---Figure 1 of Bengio et al. [1] reports the benefits of curriculum learning on several learning problems.
>
> > We agree, we do not claim that the gradient biasing and PCGrad experiments are novel.
>
> Ah this wasn't clear to me from my reading of the paper. In this case, it would be better to switch Sections 3 and 4 so that the paper's contribution comes first, and other experiments come later. Section 4 about data diversity seems to be the main story from the abstract and introduction.
>
> > We agree that a stronger understanding or real-world replication of such effects would make a much more compelling paper.
>
> I invite the authors to run these experiments. If the authors find that data diversity leads to insights in curriculum learning, a long line of work since [1] experiment on image classification and language modeling problems. Maybe there's something to learn in modern NN architectures such as transformers?
>
> [1] Bengio, Y., Louradour, J., Collobert, R., & Weston, J. (2009, June). Curriculum learning. In Proceedings of the 26th annual international conference on machine learning (pp. 41-48).

---

### Official Review · Reviewer_2yyz · 2024-11-04

**Soundness:** 3
**Presentation:** 3
**Contribution:** 2
**Rating:** 5
**Confidence:** 3

**Summary:**

The authors conduct a controlled set of experiments on three synthetic settings. They observe the loss plateau and find that by biasing the training distribution away from the test distribution to reduce data diversity at the start of training, one can accelerate training. Results from various simple and counterintuitive data interventions interventions suggest that simpler data filtering techniques can match or outperform complex optimization methods.

**Strengths:**

- The paper is overall clearly written.
- The observations are indeed interesting and surprising especially since the interventions improve training without discriminating between datapoints.

**Weaknesses:**

- From a loss landscape perspective, the initial speedup seen upon reducing data diversity suggests that the parameters are getting stuck in a local suboptimal minimum that may lie near the initialization. However, numerous methods have been developed to avoid getting stuck in such minima such as adding regularization, introducing learning rate schedulers, etc. Therefore, while reducing data diversity may seem to help speed up, it may not contribute to finding a better generalization solution at all.
- The paper claims that a purely data-driven approach can have the same speedup as optimization methods; however, low similarity and high variance are also often associated with sharp minima in the loss landscapes. Therefore, the authors could consider comparing methods like sharpness-aware minimization, momentum, etc. in this context. Moreover, PCGrad is designed for training across multiple tasks, that are expected to have potential interference. However, I wonder how scalable it would be in a single-task setup for increasing model size, even though it does improve training speed.
- While the authors mention that their synthetic settings are simple, I think finding the "optimal" amount of biasing needed for initial speedup can be quite challenging on real data, as it may require an extensive search over hyperparameters. I wonder if the authors could suggest any potential strategies for finding the optimal amount of biasing in real-world scenarios.

**Questions:**

- Fig 5, is there a reason why $\alpha=1$ performed best in all cases?
- It would be interesting to have some theoretical justification for why the interventions, especially the power law sampling, lead to increased training speed.

---

> ### Author Response · Authors · 2024-11-21
>
> Thank you for the helpful comments!
>
> > Therefore, while reducing data diversity may seem to help speed up, it may not contribute to finding a better generalization solution at all.
>
> In certain settings (i.e. factuality), training on less diverse data can result in a suboptimal solution for infinite training steps. In light of this, we introduce a curriculum learning method (detailed in Appendix A) where we slowly increase the diversity of the data over time. In all three of our settings, this curriculum uniformly dominates training on every other diversity level. Namely, for every step count, training via the curriculum achieves lower test loss compared to training on any fixed task/sample count, including the original test distribution. As such, our insight on lowering diversity can be used not only to speed up the start of training but can also be used to find the best generalizing test solution by the end of training.
>
> > The paper claims that a purely data-driven approach can have the same speedup as optimization methods; however, low similarity and high variance are also often associated with sharp minima in the loss landscapes. Therefore, the authors could consider comparing methods like sharpness-aware minimization, momentum, etc. in this context.
>
> We agree there are many optimization methods that will be able to simulate the effects of reducing data diversity. Our goal is to show that the benefits of better optimization can be curiously achieved via modulating the data instead of modifying the training algorithm. We believe this contribution will inspire future works understanding the role of data.
>
> > Moreover, PCGrad is designed for training across multiple tasks, that are expected to have potential interference. However, I wonder how scalable it would be in a single-task setup for increasing model size, even though it does improve training speed.
>
> We agree that PCGrad is not a scalable method, especially if there isn’t a natural task formulation, but the motivation for that experiment is to highlight this connection between modulating the data and the optimization algorithm. In fact, this makes our data-based interventions more attractive, as they do not require more compute or any modifications to the training algorithm.
>
> > While the authors mention that their synthetic settings are simple, I think finding the "optimal" amount of biasing needed for initial speedup can be quite challenging on real data, as it may require an extensive search over hyperparameters. I wonder if the authors could suggest any potential strategies for finding the optimal amount of biasing in real-world scenarios.
>
> Finding the optimal hyperparameters is an interesting problem for future work and we don’t expect it to be any more challenging than tuning other hyperparameters. We hope that such quantities scale naturally with model/data scale similar to learning rate [1], enabling simple tuning/prediction for practical. Additionally, it might be even easier than this, since the optimal Zipf exponent is close to 1.0 across all our settings, indicating some structure shared across different data distributions that could be leveraged to predict the optimal setting.
>
> [1] https://arxiv.org/abs/2203.03466
>
> > Fig 5, is there a reason why α=1 performed best in all cases?
>
> This is an insightful observation that we will highlight in the paper! We do not understand why this exponent for the power law performed the best and agree that it would be an interesting theoretical question to determine why this is the case. Notably, our results extend the findings of prior works [1] on the optimality of $\alpha = 1$ for the power law parameter beyond the in-context learning setting, motivating future study.
>
> [1] https://arxiv.org/abs/2205.05055

---

> ### Comment · Reviewer_2yyz · 2024-11-25
>
> I'd like to thank the authors for their responses and adding new experiment. While the authors addressed some of my concerns, a few important ones still remain unresolved:
>
> - "*Our goal is to show that the benefits of better optimization can be curiously achieved via modulating the data*"
>
> I'm not entirely convinced about this argument yet. The authors repeatedly claim that modulating the data achieves similar benefits of better optimization. Therefore, one would expect to see a direct comparison quantifying how closely this approach matches or diverges from the benefits provided by advanced optimizers. However, I couldn't find any empirical evidence to support this claim.
>
> - "*we don’t expect it (Finding the optimal hyperparameters) to be any more challenging than tuning other hyperparameters ... similar to learning rate*"
>
> I'm not sure if I understand how that might be the case. Can the authors explain this further because the optimal Zipf exponent is only observed on a synthetic setting? Therefore, I don't think finding optimal amount of biasing is as transferrable to real data as the authors mentioned. Also, it is unclear what type of biasing the authors refer to and how transferable it might be to diverse practical setups as well.
>
>
> Therefore, I'm still inclined towards 'reject' because some of the major claims are not entirely justified in the experiments and are stated as future work.

---

### Author Response · Authors · 2024-11-21

Thank you for your helpful reviews! We would like to contribute the following clarifications and experiments in light of them.

**Summary**: In this paper, we explore how reducing the diversity of the data distribution affects the model’s test loss on the original data distribution. Across multiple settings and sampling schemes, we consistently make the surprising observation that lowering the diversity of the training data distribution can lead to a faster reduction in test loss compared to training on the test distribution.

**Objective**: Our goal through this paper is not to propose a complete theory or readily practical method; instead, we hope these rigorous and broad experimental results will inspire future research in the theory and practice of how data affects optimization. We believe there are several such papers where surprising empirical observations only in synthetic settings have impacted machine learning research (some examples are (https://arxiv.org/abs/2208.01066, https://arxiv.org/abs/2306.00946, https://arxiv.org/abs/2306.15063, https://arxiv.org/abs/2301.05217, https://arxiv.org/abs/2405.15071), even without theoretical analysis.

**New Curriculum Learning Experiment**: In addition to our diversity-reducing interventions, we introduce an experiment where the data distribution follows a curriculum for our subsampling intervention (detailed in Appendix A). For these, we start training with a low number of tasks/samples, and over the course of training, increase the number of tasks/samples until reaching the original distribution. We find that for any given step count, one will achieve better test loss using the curriculum compared to training on any fixed diversity level, even as the step count increases. This shows that varying diversity can simultaneously achieve faster training and perfect test generalization.

---

### Meta-Review · Area_Chair_7hiC · 2024-12-07

**Metareview:**

This paper explores how reducing the diversity of the data distribution affects the model’s test loss on the original data distribution. Its empirical studies provide a controlled set of experiments on three synthetic settings. Observing the loss plateau, the authors draw a conclusion that by biasing the training distribution away from the test distribution to reduce data diversity at the start of training, one can accelerate training.

This is an interesting observation but there are however multiple critical concerns raised by the reviewers regarding scalability, generalizability, as well as lack of comparison with related methods such as sharpness-aware minimization. There is also the concern that this paper has very limited empirical evaluation and ultimately, lack a concrete understanding of when and how the observed phenomenon occur. Furthermore, it is also unclear to me whether the accelerated training phenomenon discussed here will necessarily improve generalizability.

The authors have made an attempt to address these concerns but the rebuttal has not addressed some concerns satisfactorily. For example, it remains unclear when and how the observed phenomenon will occur. In fact, the authors also acknowledge that they are also surprised at their own observation. In my opinion, this observation is interesting and this certainly suggests something that needs further investigation. But, unfortunately, the paper stops at reporting this observation and to me, it remains a work in progress rather than a completed work that meets the bar for acceptance. Following the rebuttal, all reviewers are still inclined towards a rejection.

**Additional Comments On Reviewer Discussion:**

The reviewers have engaged the authors during the discussion period and the concerns raised by the reviewers are valid but have not been addressed sufficiently. As a result, no reviewer has updated the rating towards acceptance. There is a clear rejection consensus and the AC recommends a rejection accordingly.

---

### Decision · Program_Chairs · 2025-01-22

Reject